# Acute Dose–Response Effectiveness of Combined Catechins and Chlorogenic Acids on Postprandial Glycemic Responses in Healthy Men: Results from Two Randomized Studies

**DOI:** 10.3390/nu15030777

**Published:** 2023-02-02

**Authors:** Aya Yanagimoto, Yuji Matsui, Tohru Yamaguchi, Shinichiro Saito, Ryuzo Hanada, Masanobu Hibi

**Affiliations:** 1Biological Science Laboratories, Kao Corporation, 2-1-3 Bunka, Sumida-ku, Tokyo 131-8501, Japan; 2Health & Wellness Products Research Laboratories, Kao Corporation, 2-1-3 Bunka, Sumida-ku, Tokyo 131-8501, Japan; 3SOUSEIKAI Sumida Hospital, 1-29-1 Honjo, Sumida-ku, Tokyo 130-0004, Japan

**Keywords:** coffee, cookie meal test, glucagon-like peptide-1, glucose-dependent insulinotropic polypeptide, green tea

## Abstract

Epidemiologic studies show that the risk of diabetes can be reduced by ingesting green tea or coffee. Previous studies have shown that simultaneously taking green tea catechins (GTC) and coffee chlorogenic acid (CCA) alters postprandial gastrointestinal hormones secretion and improves insulin sensitivity. However, there is no evidence on the acute effects of GTC and CCA on incretin and blood glucose, and on the respective dose of polyphenols. In this randomized, double-blind, placebo-controlled crossover study, we examined the effective dose of GTC and CCA on postprandial glucose, insulin, and incretin responses to a high-fat and high-carbohydrate cookie meal containing 75 g of glucose in healthy men. Study 1 (*n* = 18) evaluated two doses of GTC (270 or 540 mg) containing a fixed dose of CCA (270 mg) with 113 mg of caffeine and a placebo (0 mg GTC and 0 mg CCA) with 112 mg of caffeine. Study 2 (*n* = 18) evaluated two doses of CCA (150 or 300 mg) containing a fixed dose of GTC (540 mg) and a placebo with 99 mg of caffeine. The single combined ingestion of GTC and CCA significantly altered the incretin response and suppressed glucose and insulin levels. These findings suggest that the effective minimum dose is 540 mg of GTC and 150 mg of CCA.

## 1. Introduction

Type 2 diabetes (T2D), a serious chronic disease caused by insulin resistance and impaired insulin secretion [1], is a global health and wellness issue [1,2,3]. Persistent hyperglycemia leads to insulin resistance. Blood glucose levels are strictly regulated by insulin, glucagon, cortisol, catecholamines (adrenal corticosteroids), growth hormone, cortisol, and aldosterone and are also influenced by stress. Incretins, such as glucose-dependent insulinotropic polypeptide (GIP) and glucagon-like peptide-1 (GLP-1) secreted from the small intestine in response to dietary glucose and lipids, are required for insulin secretion from pancreatic β-cells. GLP-1 enhances glucose uptake in the liver and muscle and improves systemic insulin sensitivity [4,5]. GIP promotes fat accumulation via glucose and free fatty acid uptake in adipocytes [6]. Healthy eating habits from childhood are a factor in preventing the development of T2D. Improving insulin resistance and controlling blood glucose levels through increased physical activity may also prevent T2D. The effectiveness of non-pharmacologic therapies, however, including food-derived functional ingredients, for improving insulin resistance has not been established. 

Green tea and coffee, which contain abundant polyphenols such as green tea catechins (GTC) and coffee chlorogenic acids (CCA), respectively, are the most commonly consumed beverages in the world. Consumption of GTC or CCA may reduce the risk of developing T2D and other metabolic diseases [7,8,9,10,11]. A previous study in healthy men demonstrated that consuming green tea extract containing GTC ameliorates insulin sensitivity in healthy individuals [12]. Chronic consumption of beverages containing 582 mg of GTC improved hemoglobin A_1C_ (HbA_1c_) levels in participants taking medicines that stimulate insulin secretion [13]. In individuals with impaired glucose tolerance, consumption of 400 mg of CCA for 3 months significantly reduced fasting blood glucose levels and the insulinogenic index on the basis of fasting and postprandial blood glucose and insulin levels [14]. Acute ingestion of CCA suppresses postprandial blood glucose levels [15] and promotes GLP-1 secretion [16]. These findings suggest that ingesting a combination of GTC and CCA may further enhance their effects to improve glucose metabolism. In addition to the control of glucose and insulin responses, regulation of incretins is a potential therapeutic treatment option for insulin resistance [17,18]. Few studies, however, have evaluated the combined effects of GTC and CCA on postprandial glucose metabolism and gastrointestinal hormone secretion in humans [19]. In addition, the acute effects of simultaneous consumption of GTC and CCA have not been clarified, and the mechanisms by which these polyphenols affect glucose metabolism are unclear. To the best of our knowledge, there are no studies evaluating human glucose metabolism using different GTC and CCA doses. Clarifying the minimum effective doses of GTC and CCA to better understand the relationship could provide valuable information for future study designs. 

In the present study, we evaluated the acute effects of four combinations of GTC and CCA concentrations on postprandial glucose metabolism to determine the dose–response relationship after ingestion in healthy, non-diabetic individuals. The primary endpoint was the minimum dose required for a postprandial GLP-1 response, and the secondary endpoints were postprandial blood glucose, insulin, and GIP responses. 

## 2. Materials and Methods

### 2.1. Study Design

Effective combined doses of GTC and CCA were investigated in two human clinical trials conducted at Kao Corporation (Tokyo, Japan) and Sumida Hospital (Tokyo, Japan). Study 1 investigated the effective dose of GTC with a fixed dose of CCA, and Study 2 investigated the effective dose of CCA with a fixed dose of GTC. 

Both acute ingestion studies were conducted, in which the three test beverages were each consumed immediately after ingestion of the cookie test meal, and the effects on postprandial glucose metabolism following 4 h were evaluated. A one-week washout was provided between the consumption of each of the three test beverages in a randomized crossover manner. In each study, the participants consumed test beverages containing two different dose combinations of GTC and CCA or a placebo. All of the participants in both studies were enrolled and randomly allocated a number using a computer-generated stratified randomization method to test the sequence groups. The sequence group allocation was concealed among the participants, physicians, and outcome assessors from screening to finalizing the dataset.

Study 1 and Study 2 participants were recruited separately, each with another 18 participants. Study 1 was conducted from October 2017 to February 2018. Each participant orally ingested a test beverage containing a fixed dose of CCA (270 mg) combined with either a minimum dose of GTC (540 mg: reported as the minimum effective dose for metabolic disorders [20]) or a half dose of GTC (270 mg) or a placebo beverage containing 0 mg GTC and 0 mg CCA. All of the test beverages contained 112–113 mg of caffeine. Study 2 was conducted from August 2019 to October 2019. Each participant orally ingested a beverage containing a fixed dose of GTC (540 mg) combined with either a minimum dose of CCA (300 mg: reported as the minimum effective dose for metabolic disorders [21]), a half dose of CCA (150 mg), or a placebo beverage containing 0 mg GTC and 0 mg CCA. All of the test beverages contained 99 mg caffeine. The two studies were conducted in accordance with the Declaration of Helsinki, and the protocols were reviewed and approved by the ethics committees of Kao Corporation (No. T232-190615, 30 July 2019; Tokyo, Japan) and Ueno-Asagao Clinic (No.2019-19; 7 August 2019; Tokyo, Japan). All of the participants received a full verbal and written explanation of the study and provided written informed consent prior to registration. The respective trials were registered with the University Hospital Medical Information Network (UMIN; http://www.umin.ac.jp/ (accessed on 2 January 2023); Registration No. UMIN000047544 for Study 1, UMIN000037738 for Study 2).

### 2.2. Participants

In Study 1, potential participants were recruited from among healthy male volunteers living in the Tokyo metropolitan area in October 2017. Eighteen healthy volunteers (mean ± SD: age, 41 ± 9 years; body mass index [BMI]: 22.6 ± 1.8 kg/m^2^; fasting glucose, 86 ± 5 mg/dL) with normal blood glucose levels (fasting blood glucose, ≤125 mg/dL) were recruited to participate in this study. The exclusion criteria were a history of diabetes or cardiovascular disease, hypertension, hypercholesterolemia, hypertriglyceridemia, and hepatic, renal, or gastrointestinal diseases; smoking habit; excessive alcohol consumption (≥30 g/day); allergies to the ingredients in the test food; working as a shift worker; or otherwise determined to be unqualified by the physician in charge. In Study 2, potential participants were recruited from among healthy male volunteers living in the Tokyo metropolitan area in September 2019. Eighteen healthy male volunteers (mean ± SD; age: 47 ± 11 years; BMI: 23.2 ± 2.9 kg/m^2^; fasting glucose: 103 ± 7 mg/dL) with blood glucose levels ranging from normal to impaired glucose tolerance (fasting glucose 95–125 mg/dL) were recruited according to the same criteria as in Study 1. The sample size for each study was estimated according to the results of a preliminary study and calculated to be 18, assuming a significance level of 5% and a statistical power of 80%. In the preliminary study, the effect size of the GLP-1 area under the curve (AUC) was 7.7 mg/dL·4 h with a standard deviation (SD) of 11.0 mg/dL·4 h.

### 2.3. Experimental Procedures

In both studies, after a 1-week run-in period, measurements for the acute ingestion test were obtained after the participants consumed one of the three test beverages. The day before the measurements, the participants were instructed to consume pre-packaged meals and to go to bed before midnight without consuming any food or beverages other than water after 9:00 p.m. In Study 1, the total energy content of the pre-packaged meals the day before the test was 2412 kcal/day, 14 E% from protein, 25 E% from fat, 61 E% from carbohydrates; 25 E% for breakfast, 24 E% for lunch, 21 E% for snacking, and 30 E% for dinner. In Study 2, the total energy content of the pre-packaged meals the day before the test was 2166 kcal/day, 14 E% from protein, 25 E% from fat, 61 E% from carbohydrates; 34 E% for breakfast, 32 E% for lunch, and 34 E% for dinner. The height, weight, body fat (body fat scale by bioelectrical impedance analysis, model TF-780 in Study 1 and model DC-320 in Study 2, Tanita Corp., Tokyo, Japan), waist circumference, and blood pressure (digital blood pressure monitor, model HEM-1000, Omron, Kyoto, Japan in Study 1 and model TM-2571, A&D Co., Ltd., Tokyo, Japan in Study 2) were measured for each participant. The participants consumed a high-fat and high-carbohydrate cookie meal which consisted of 12 test-specific cookies (meal test S, Saraya Co., Ltd., Osaka, Japan) over a 15-min period, and then the subjects consumed the test beverage over a 10-min period. The cookie meal contained flour, butter, maltose, chicken eggs, and baking powder and had an energy content of 592 kcal (6 E% from protein, 43 E% from fat, and 50 E% from carbohydrates). Venous blood samples were obtained during fasting (0 h) and at 0.5, 1, 1.5, 2, and 4 h after cookie meal consumption in Study 1 and during fasting (0 h) and at 0.5, 1, 1.5, 1.75, 2, 3, and 4 h after cookie meal consumption in Study 2. The primary outcome measurement was postprandial GLP-1 concentration, and the key secondary outcomes were postprandial blood glucose, insulin, and GIP concentrations. 

### 2.4. Test Beverages

The test beverage compositions are summarized in Table 1. The test beverages were 350 mL tea-flavored beverages in both Studies 1 and 2. All of the test beverages were prepared by the Kao Corporation (Tokyo, Japan). The amount of GTC and caffeine in the test beverages was measured by high-performance liquid chromatography using an L-column ODS (4.6 mm diameter × 250 mm length; Chemicals Evaluation and Research Institute, Tokyo, Japan) and CCA using a Cadenza CD C18 column (4.6 mm diameter × 150 mm length; Intact, Kyoto, Japan). In Study 1, the GTC comprised catechin, epicatechin, gallocatechin, epigallocatechin, catechin gallate, epicatechin gallate, gallocatechin gallate, and epigallocatechin gallate in green tea extracts. The CCA comprised 3-, 4-, 5-caffeoylquinic acids and 3-, 4-, 5-feruloylquinic acids in coffee bean extracts. Each group in Study 1 is referred to hereafter as follows: Placebo (0 mg GTC, 0 mg CCA, with 112 mg), Dose A (270 mg GTC, 270 mg CCA, with 113 mg caffeine), and Dose B (540 mg GTC, 270 mg CCA, with 113 mg caffeine). Each group in Study 2 is referred to hereafter as follows: Placebo (0 mg GTC, 0 mg CCA, with 99 mg), Dose C (540 mg GTC, 150 mg CCA, with 99 mg caffeine), and Dose D (540 mg GTC, 300 mg CCA, with 99 mg caffeine). All of the test beverages, including the placebo, were prepared to be indistinguishable in appearance, taste, and smell.

### 2.5. Biochemical Analysis

The plasma and serum samples were obtained from the participants in a fasting state and after cookie meal ingestion, snap-frozen using liquid nitrogen, and immediately stored at −80 °C until the analysis was performed. Glucagon, active GLP-1, and total GIP levels were measured by enzyme-linked immunosorbent assay (glucagon: Mercodia, Uppsala, Sweden; active GLP-1: Immune Biology Laboratories, Gunma, Japan; total GIP: Merck Millipore, Darmstadt, Germany). All of the other items were analyzed at LSI Medience, Co., Ltd. (Tokyo, Japan). Blood glucose was measured by an enzymatic method and insulin was measured by chemiluminescence immunoassay (CLIA).

### 2.6. Statistical Analysis

Microsoft Excel 14.0, SPSS version 28.0 (IBM Corp., Armonk, NY, USA) and SAS version 9.4 (SAS Institute Inc., Cary, NC, USA) were used for the statistical analysis. Data are expressed as mean ± standard error of the mean (SEM). The AUCs were calculated using the trapezoidal rule, maximum concentration (C_max_), and effect size, and the 95% confidence interval (CI) was calculated based on measurements at each time-point for the primary and secondary outcome measures. The following statistical analyses were performed among the three test beverage conditions. The values at each time-point were determined for each participant and entered into a linear mixed-effmixed-effect model with the treatment, time, and treatment-by-time interaction as the fixed effects. The significance of the total AUC (tAUC) and incremental AUC (iAUC) was analyzed using a mixed-effect model with adjustment of the effects of period and order by the “SAS GLIMMIX procedure” which allows statistical models to be fitted when responses are not necessarily normally distributed, and the effect of each test beverage was estimated with the “ESTIMATE” statement. Significance was estimated using empirical variance (“EMPIRICAL” option). An unstructured (“type = UN”) variance-covariance matrix in the data with repeated measures was assumed. The Tukey–Kramer method was used to adjust for statistical multiplicity. In addition, stratified and correlation analyses were performed according to subject characteristics. In all of the analyses, the significance level was set at 5%.

## 3. Results

### 3.1. Participants

All of the participants completed Study 1. In Study 2, two of the participants declined to continue the test for personal reasons; therefore, 16 participants completed the study. No participants were excluded from the analysis according to the study protocols; data at the end of the study for the 18 and 16 participants who completed the studies were determined and an intention to treat analysis was performed for each study. The anthropometric measurements and the blood glucose and insulin levels at baseline in Studies 1 and 2 are shown in Table 2. No adverse events caused by the test beverages were noted. The demographic characteristics of the participants enrolled in Studies 1 and 2 were generally similar in terms of age, weight, height, and BMI.

### 3.2. Blood Glucose and Insulin

The fasting glucose levels of all of the participants were in the normal to borderline range (fasting blood glucose level ≤ 125 mg/dL) during all of the periods in Studies 1 and 2. Figure 1 shows changes in blood glucose and insulin concentrations before and after the cookie meal, and Table 3 (Study 1) and Table 4 (Study 2) show estimated values of the tAUC adjusted for the effects of the order and period. In Study 1, comparisons of the three conditions using linear mixed-effect models revealed a treatment effect for the change in blood glucose and serum insulin but no treatment by time interaction effect. The results of the mixed-effect model analysis of the comparison of the three groups revealed a difference in the tAUC for blood glucose, but not for insulin. Furthermore, the tAUC for blood glucose was significantly lower in the Dose B group than in the Placebo group. 

In Study 2, comparisons of the three conditions using linear mixed-effect models revealed a treatment effect for the change in blood glucose and serum insulin. The comparison of the three groups analyzed by the mixed-effect model revealed a significant difference in the tAUC among the three groups in both blood glucose and serum insulin. Furthermore, the tAUC for blood glucose was significantly lower in the Dose C group and the Dose D group than in the Placebo group. Furthermore, the Dose D group had a significantly lower tAUC value than the Placebo group.

### 3.3. Incretins

Figure 2 shows changes in the plasma GLP-1 and GIP concentrations before and after the cookie meal, and Table 3 (Study 1) and Table 4 (Study 2) show estimated values of the iAUC adjusted for the effects of the order and period. We assessed the differences in the iAUC for incretin responses because the fasting values fluctuated between each test. The tAUC was used for the blood glucose and serum insulin levels because the fasting values between each test were strictly regulated within individuals, and thus, postprandial values could be less than 0.

In Study 1, the change in the postprandial GLP-1 and GIP levels revealed the main treatment effect, but no treatment by time interaction effect in a mixed-effect model of the measurement data. The results of the mixed-effect model analysis of comparison of the three groups showed a non-significant tendency toward a difference in the iAUC for plasma GLP-1 and a significant difference for plasma GIP. Furthermore, the iAUC for GLP-1 in the Dose A group exhibited an increasing trend, and the Dose B group had a significantly higher iAUC value than the Placebo group. On the other hand, the iAUC for plasma GIP was significantly lower in both the Dose A and Dose B groups than in the Placebo group.

In Study 2, the change in the postprandial GLP-1 and GIP showed the main treatment effect, but no treatment-by-time interaction effect in the mixed-effect model. A comparison of the three groups analyzed by the mixed-effect model revealed a significant difference in the tAUC of both the plasma GLP-1 and GIP levels among the three groups. The tAUC for GLP-1 was significantly higher in both the Dose C and Dose D groups than in the Placebo group, while the tAUC for GIP was significantly lower in both the Dose C group and Dose D group than in the Placebo group.

## 4. Discussion

This study was performed to estimate the effective dose for a single ingestion of both GTC and CCA on postprandial glucose metabolism, including incretin responses. To our knowledge, this study is the first to assess these conditions, and our findings suggest that a single intake of a combination of varying doses of GTC and CCA significantly improved postprandial hyperglycemia, increased GLP-1, and decreased GIP secretion after consumption of a high-fat and high-carbohydrate cookie meal containing 75 g of glucose. In addition, a comparison of the test beverages with different concentrations of GTC or CCA revealed the minimum effective dose affecting the glycemic and incretin responses. Specifically, compared with the Placebo, the Dose B group (540 mg GTC, 270 mg CCA, with 113 mg caffeine), but not the Dose A group (270 mg GTC, 270 mg CCA, with 113 mg caffeine), maintained the treatment efficacy in Study 1, while both the Dose C group (540 mg GTC, 150 mg CCA, with 99 mg caffeine) and the Dose D group (540 mg GTC, 300 mg CCA, with 99 mg caffeine) maintained the treatment efficacy in Study 2. Based on the results of the present studies, the minimum effective doses of GTC and CCA for promoting the secretion of GLP-1 and GIP are 540 mg and 150 mg, respectively.

Previous studies in humans have not find postprandial enhancement of GLP-1 secretion by GTC intake alone. In contrast, studies investigating the effects of coffee or CCA alone revealed improvements in postprandial hyperglycemia and increased GLP-1 secretion [16,22]. Comparison of the present results with these studies is difficult, due to differences in the content of the test meals and the duration of the test beverage consumption period. Fujii et al. [23] reported that GLP-1 secretion is enhanced in a CCA dose-dependent manner in NCI-H716 cells. This may be related to the combined use with GTC as opposed to CCA alone. It should be noted that the CCA dose in the in vitro experiments cannot be directly compared to that in the human study, nor do the results of in vitro experiments reflect the effects of various dietary nutrients ingested simultaneously in vivo.

In the present study, acute incretin responses were observed following the ingestion of test beverages with high GTC and CCA concentrations, suggesting that GTC and CCA are involved in the mechanisms that promote GLP-1 and inhibit GIP secretions. Various phytochemicals, such as several types of flavonoids, black-tea polyphenols, oolong tea polymerized polyphenols, and potato polyphenolic compounds, inhibit the function of nutrient-digesting enzymes [24,25,26,27]. Blood incretin concentrations are affected by the intake of glycolytic and lipolytic enzyme inhibitors [28,29,30]. GLP-1 is rapidly secreted 15 to 30 min after a meal, mainly from L cells in the lower small intestine, and peaks at 90 to 120 min. GLP-1 secretion is presumed to be caused by stimulation of the vagus nerve or other hormones, whereas GIP secretion is thought to follow direct stimulation within the small intestinal lumen [4]. The nutrition factors reported to stimulate GLP-1 and GIP secretion include sodium–glucose cotransporter protein-mediated glucose and others and various fatty acids via G protein-coupled receptors [31,32,33]. It was suggested that the combined intake of GTC and CCA inhibits the digestion and absorption of nutrients such as carbohydrates and lipids, promotes increased secretion of GLP-1, and inhibits increased secretion of GIP, rather than directly stimulating K and L cells in the small intestine. In other words, the digestion of carbohydrates or lipids by amylase/glucosidase or lipase may be delayed by decreased stimulation in the upper part of the small intestine and increased stimulation in the lower small intestine. Of note, the consumption of the test beverages with high concentrations of both GTC and CCA decreased postprandial blood glucose levels, suggesting that polyphenols inhibit the absorption of dietary glucose into the blood. The difference in the effect on postprandial blood glucose levels between Studies 1 and 2 may be due to differences in the participants: in Study 1, no specific blood glucose levels were set as exclusion criteria, while in Study 2, blood glucose levels ranged from normal to impaired glucose tolerance (95–125 mg/dL); the mean fasting blood glucose level of the participants in Study 1 was 86 ± 5 mg/dL and that of the participants in Study 2 was 103 ± 7 mg/dL. Therefore, a more remarkable effect of the combined ingestion of GTC and CCA was observed in the participants of Study 2, who had higher blood glucose levels. Similarly, an increase in GLP-1 secretion and a decrease in GIP secretion is observed with α-glucosidase inhibitors [34] in drugs and apple juice [35], suggesting that the absorption site of carbohydrates is shifted to the lower part of the small intestine. Other studies have reported postprandial increases in GLP-1 following single and continuous (90 days) ingestion of green-plant membranes [36]. The mechanism of this GLP-1 secretion enhancement is considered to be a direct effect on endocrine cells in the small intestine via prolonged digestion and absorption of dietary lipids or an indirect effect mediated by neural signals such as cholecystokinin, but this has not yet been clarified.

This study has some potential limitations. The participants were healthy (normal) or had impaired glucose tolerance but were non-diabetic individuals. Therefore, it is unclear whether a similar efficacy on GLP-1 secretion would occur in individuals with T2D. This study was conducted on men only, and caution should be exercised in making general inferences. We hypothesized that a woman’s menstrual cycle could affect her blood glucose levels because of the 1-week washout period between each test. Caffeine is a major active component of green tea and coffee, but the synergistic effects of caffeine with catechins and chlorogenic acid were not considered in this study. While caffeine consumption may improve glucose tolerance, its effect on glucose levels varies among studies [37]. On the other hand, several studies report that caffeine consumption increases insulin secretion, but does not necessarily improve glucose levels in oral glucose tolerance tests [38,39,40], suggesting that caffeine influences insulin clearance. In the present study, the amount of caffeine was adjusted to be the same in the placebo and GTC plus CCA test beverages, but the possibility that caffeine contributed to the observed effects cannot be ruled out. Furthermore, due to differences in estimated energy intake the day before the test between Studies 1 and 2, fasting blood glucose levels on the day of the test may have been higher in Study 1. Therefore, although the same diet was consumed in each study, the findings should be cautiously interpreted.

In conclusion, the minimum dose for the effect of a single oral intake of GTC and CCA on incretin levels in healthy men after consuming the high-fat, high-carbohydrate cookie meal containing 75 g of glucose was found to be 540 mg of GTC and 150 mg of CCA. Furthermore, at the same time, 540 mg of GTC and 150 mg of CCA inhibited the increase in blood glucose levels after ingestion of the high-fat and high-carbohydrate cookie meal.

## Figures and Tables

**Figure 1 nutrients-15-00777-f001:**
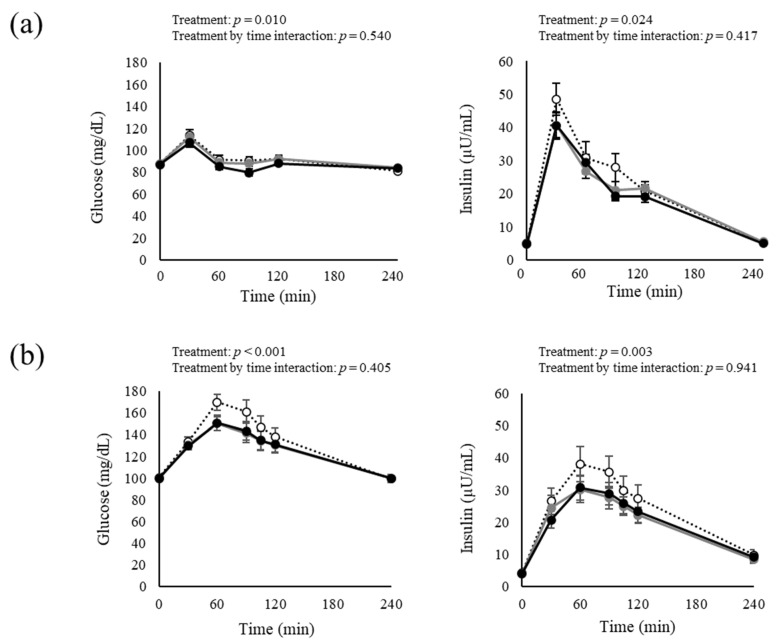
Changes in blood test results at fasting and after cookie meal consumption for Study 1 (**a**) and Study 2 (**b**). Data are expressed as mean ± SEM. (**a**) White circles indicate the Placebo. Gray and black circles indicate Dose A (270 mg GTC, 270 mg CCA, with 113 mg caffeine) and Dose B (540 mg GTC, 270 mg CCA, with 113 mg caffeine), respectively. Treatment main effect and treatment × time interactions were analyzed by linear mixed-effect models (*n* = 18). (**b**) White circles indicate the Placebo group. Gray and black circles indicate the Dose C group (540 mg GTC, 150 mg CCA, with 99 mg caffeine) and the Dose D group (540 mg GTC, 300 mg CCA, with 99 mg caffeine), respectively. Treatment main effects and treatment × time interactions were analyzed by linear mixed-effect models (*n* = 18; Placebo, *n* = 17; Dose C, Dose D). Abbreviations: CCA, coffee chlorogenic acid and GTC, green tea catechin.

**Figure 2 nutrients-15-00777-f002:**
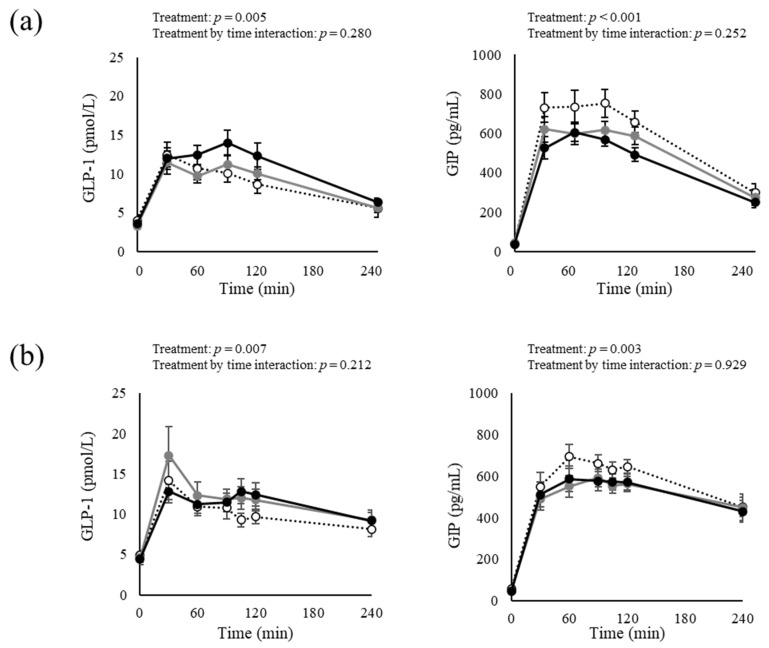
Changes in blood test results at fasting and after cookie meal consumption for Study 1 (**a**) and Study 2 (**b**). Data are expressed as mean ± SEM. (**a**) White circles indicate the Placebo. Gray and black circles indicate Dose A (270 mg GTC, 270 mg CCA, with 113 mg caffeine) and Dose B (540 mg GTC, 270 mg CCA, with 113 mg caffeine), respectively. Treatment main effects and treatment × time interactions were analyzed by linear mixed-effect models (*n* = 18). (**b**) White circles indicate the Placebo. Gray and black circles indicate Dose C (540 mg GTC, 150 mg CCA, with 99 mg caffeine), and Dose D (540 mg GTC, 300 mg CCA, with 99 mg caffeine), respectively. Treatment main effects and treatment × time interactions were analyzed by linear mixed-effect models (*n* = 18; Placebo, *n* = 17; Dose C and Dose D). Abbreviations: CCA, coffee chlorogenic acid; GIP, glucose-dependent insulinotropic polypeptide; GLP-1, glucagon-like peptide-1; and GTC, green tea catechin.

**Table 1 nutrients-15-00777-t001:** Test beverage compositions.

	Study 1			Study 2		
	Placebo	Dose A	Dose B	Placebo	Dose C	Dose D
Catechin (mg)	0	14	27	0	30	30
Epicatechin (mg)	0	15	30	0	32	32
Gallocatechin (mg)	0	60	119	0	119	119
Epigallocatechin (mg)	0	54	108	0	103	103
Catechin gallate (mg)	0	11	23	0	23	23
Epicatechin gallate (mg)	0	17	33	0	37	37
Gallocatechin gallate (mg)	0	49	99	0	93	93
Epigallocatechin gallate (mg)	0	50	101	0	105	105
Total catechins (mg)	0	270	540	0	540	540
3-caffeoylquinic acid (mg)	0	82	83	0	46	92
4-caffeoylquinic acid (mg)	0	62	62	0	34	68
5-caffeoylquinic acid (mg)	0	73	74	0	41	82
3-feruloylquinic acid (mg)	0	19	19	0	11	23
4-feruloylquinic acid (mg)	0	15	15	0	16	16
5-feruloylquinic acid (mg)	0	18	17	0	10	19
Total chlorogenic acids (mg)	0	270	270	0	150	300
Caffeine (mg)	112	113	113	99	99	99

These values were measured by high-performance liquid chromatography. The volume of the tea-flavored test beverage was 350 mL. Abbreviations: CCA, coffee chlorogenic acid and GTC, green tea catechin.

**Table 2 nutrients-15-00777-t002:** Physical characteristics, fasting blood glucose, and insulin at baseline.

		Study 1		Study 2	
		Mean ± SD	Range	Mean ± SD	Range
Age	(y)	41 ± 9	30–59	47 ± 11	33–63
Weight	(kg)	68.5 ± 1.7	55.8–88.8	69.0 ± 10.1	49.8–91.0
BMI	(kg/m^2^)	22.6 ± 1.8	19.5–26.5	23.2 ± 2.9	18.5–27.4
Body Fat	(%)	19.5 ± 3.5	14.1–26.2	21.6 ± 5.1	13.1–31.0
HbA1c	(%)	5.3 ± 0.2	4.8–5.6	5.4 ± 0.3	4.9–5.9
FBG	(mg/dL)	86 ± 5	79–98	103 ± 7	95–115
Insulin	(µU/mL)	5.0 ± 1.7	2.8–7.9	3.7 ± 1.4	1.6–6.4

Data are expressed as mean ± SD (Study 1: *n* = 18; Study 2: *n* = 18). Abbreviations: SD: standard deviation, BMI: body mass index, HbA1c: hemoglobin A1c, and FBG: fasting blood glucose.

**Table 3 nutrients-15-00777-t003:** AUC for blood-glucose-related indices and incretin up to 4 h after cookie meal ingestion in Study 1.

		Placebo	Dose A vs. Placebo	Dose B vs. Placebo	*p* Value (3 Group)
Glucose	tAUC mg/dL for 4 h	370 ± 7	363 ± 5	350 ± 6 *	0.058
Insulin	tAUC µU/mL for 4 h	83.1 ± 6.4	79.5 ± 5.8	76.3 ± 5.3	0.152
GLP-1	iAUC pmol/L for 4 h	17.7 ± 2.8	23.0 ± 2.5	26.6 ± 3.2 *	0.065
GIP	iAUC pg/mL for 4 h	2243.7 ± 107.3	1659.1 ± 82.7 ***	1518.8 ± 92.6 ***	<0.001 ***

Data are expressed as estimate ± SEM. Comparisons of the three groups and tests of the differences between the groups were analyzed by a mixed-effect model. Multiplicity was adjusted using Tukey–Kramer’s test (*n* = 18, significant difference compared to the Placebo; * *p* < 0.05, *** *p* < 0.001). Dose A refers to the test beverage containing 270 mg of GTC, 270 mg of CCA, and 113 mg caffeine; Dose B refers to the test beverage containing 540 mg of GTC, 270 mg of CCA, and 113 mg caffeine in Study 1. Abbreviations: CCA, coffee chlorogenic acid; GIP, glucose-dependent insulinotropic polypeptide; GLP-1, glucagon-like peptide-1; and GTC, green tea catechin.

**Table 4 nutrients-15-00777-t004:** AUC for blood-glucose-related indices and incretin up to 4 h after cookie meal ingestion in Study 2.

		PLA	Dose C vs. Placebo	Dose D vs. Placebo	*p* Value (3 Group)
Glucose	tAUC mg/dL for 4 h	535 ± 19	490 ± 15 **	497 ± 18 **	0.001 **
Insulin	tAUC µU/mL for 4 h	94.5 ± 9.5	78.8 ± 5.8	77.4 ± 7.5 *	0.048 *
GLP-1	iAUC pmol/L for 4 h	21.3 ± 2.1	26.1 ± 2.7 *	26.8 ± 2.8 *	0.011 *
GIP	iAUC pg/mL for 4 h	1970.2 ± 115.4	1818.2 ± 94.0	1800.3 ± 103.1 **	0.004 **

Data are expressed as estimate ± SEM. Comparison of the three groups and tests of the differences between the groups were analyzed by a mixed-effect model. Multiplicity was adjusted using Tukey–Kramer’s test (*n* = 18; Placebo, *n* = 17; Dose C and Dose D, significant difference compared to the Placebo; * *p* < 0.05, ** *p* < 0.01). Dose C refers to the test beverage containing 540 mg GTC, 150 mg CCA, and 99 mg caffeine; Dose D refers to the test beverage containing 540 mg GTC, 300 mg CCA, and 99 mg caffeine in Study 2. Abbreviations: CCA, coffee chlorogenic acid; GIP, glucose-dependent insulinotropic polypeptide; GLP-1, glucagon-like peptide-1; GTC, and green tea catechin.

## Data Availability

Not applicable.

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
