# Peer review of "Acute Dose–Response Effectiveness of Combined Catechins and Chlorogenic Acids on Postprandial Glycemic Responses in Healthy Men: Results from Two Randomized Studies"

_nutrients, 2023, doi:10.3390/nu15030777_

Round 1
Reviewer 1 Report (New Reviewer)
The present manuscript reports on two acute interventions assessing the effect of combination of green tea catechins (GTC) and coffee chlorogenic acids (CCA) at different concentrations plus caffeine on the post-prandial glucose and insulin levels as well as on incretin hormones (GLP-1 and GIP) in healthy male volunteers with normal basal glucose levels. These authors have recently published an article assessing the effect of a 3-week consumption of a similar combination of GTC+CCA (at a different dose from the ones here assayed) on glucose, insulin and incretins in healthy men (Ref 19 in the manuscript)
In spite of this, the manuscript holds some novelty. However, it only assesses the four referred parameters (glucose, insulin, GLP-1 and GIP), which limits its potential relevance and interest. In addition, the manuscript lacks clarity, missing important descriptions, with contradictions in other key aspects. The abstract is very confusing, the introduction is not adequate, detailing some studies among the many reporting the effects of tea or coffee phenolic compounds on glucose homeostasis and insulin resistance, not stablishing a proper hypothesis and failing to convey the real interest and objective of the studies. Similarly, the discussion is not focused and needs to be thoroughly revised.
All these critical aspects need to be corrected before considering the manuscript suitable for publication.
Of crucial importance, the actual study design is not clear, since there is no information on when the test beverages were consumed by the volunteers. Together with the high-fat, high-carbohydrate cookies? Before? After? If so, with how much time difference? How much time did the volunteers have to ingest the cookie (and beverage)? This is important since incretins’ response occurs fast after food consumption and therefore the amount of food and the time available for its intake must be standardized in both studies. Could this have affected the differences observed in study 2 that were not found in study 1? Also, please indicate how much apart was each intervention (you indicate that after a 1-week run-in period in line 126).
Other important information missing in the manuscript is the inclusion criteria or which parameter was used to estimate the sample size, among others. At this respect, it is not sufficient to indicate that sample size was estimated according to a non-referred preliminary study (line 119). Was post-prandial GLP-1 concentration (primary outcome) used for calculation of the sample size? Also, please justify the selection of GLP-1 levels as primary outcome.
The fact that both studies were carried out in healthy men is an important drawback, as authors already acknowledged identifying it as a limitation. They should justify their selection of the study population.
Among the contradictions found in the manuscript outstand the caffeine content of the test and placebo beverages. There are no two statements either in the abstract, experimental section, results (table 1) or discussion with the same information. Authors must clearly state how much caffeine contained each beverage in both studies. This incongruencies diminishes the confidence of readers on the manuscript.
Authors indicate (lines 342-346) that the effect of the combined intake of GTC and CCA on incretins suggests that “the combination stimulates the digestion and absorption of nutrients, such as carbohydrates and lipids”. This is against the well-known effect of phenolic compounds inhibiting digestive enzymes and interfering on nutrients absorption. It is an affirmation not substantiated by any mechanistic study nor even a simple in vitro digestion of, for example, the test cookie with the combined GTC+CCA. But this suggestion is not even sustained by the results reported in the manuscript, since a reduced postprandial glucose levels were observed after the intake of the GTC+CCA. Again, important contradictions showing a poorly mediated discussion that weakens the relevance and potential impact of the manuscript.
Other comments
In lines 84-90 it is said that “an optimal” dose of GTC and CCA were used, “reported as the minimum effective dose for metabolic disorders”. Is really a “minimum dose” the “optimal”? This is confusing.
The use of terms like test meal, test cookies, test beverages throughout the manuscript is confusing. Please standardize the terminology. Similarly, the terms high GTC/high CCA, low GTC/high CCA, high GTC/low CCA, high GTC/high CCA are not readers’ friendly. Could you not find some other easier way to refer to the different groups?
In line 101 you mention that study diagrams are shown in Figures 1 and 2, but these figures correspond to results.
In line 153 you indicate that the test beverages were analyzed by HPLC. Please briefly describe the analytical conditions. How was caffeine analyzed? Next, you indicate that “In study 1, the GTC comprised catechin…”. Actually, in both studies the GTC and CCA had identical phenolic composition, only varying the final amounts of polyphenols. Most importantly, please clarify wat was used to prepare the GTC+CCA beverages, whether you used tea and coffee phenolic extracts or pure phenolic compounds. If using extracts, please describe how the extracts were obtained or, if using pure standards, indicate their precedence.
Please indicate in the experimental section how many high-fat, high-carbohydrate cookies (or grams) were consumed in each test day and specify the amount of glucose they provide (this is only mentioned in the abstract and discussion). Also, specify the volume of beverage consumed (this information is only included as a footnote in table 1).
There is no information on how glucose and insulin were measured. Please include in section 2.5.
In line 234 you indicate that the high GTC/low CCA group exhibited a decreasing trend in the tAUC for serum insulin; a p=0.048 shows statistical significance, not “a trend”. On the contrary, in Table 3 p values of 0.058 and 0.065 for glucose and GLP-1, respectively, are not statistically significant, and therefore no asterisk should be included nor mention of significant differences compared to placebo.
In paragraph in lines 284-290 all tAUC should be changed to iAUC.
In line 379 authors refer to differences in estimated energy expenditure the day before the test between studies 1 and 2 as a potential explanation for the differences observed in fasting blood glucose levels. How the energy expenditure was measured/calculated must be described in the experimental section and results included in the manuscript to sustain this explanation.
Finally, the post-prandial increase in blood glucose and insulin was not “suppressed” after intake of the GTC+CCA beverages, as concluded in line 384. Please rephrase.
Author Response
Response to Reviewer Comments
January 23, 2023
Nutrients
Dear Reviewers and Editors,
We would like to thank you very much for reviewing and commenting on our manuscript. We have carefully considered all your comments and made major revisions to the manuscript as a result. Below, we present our responses to each of the comments. We have significantly revised this manuscript in accordance with the your comments. We believe that these changes have greatly improved our paper. We hope that our revised version satisfies the reviewers and editors.
-------------------------
Reviwer comments:
The abstract is very confusing, the introduction is not adequate, detailing some studies among the many reporting the effects of tea or coffee phenolic compounds on glucose homeostasis and insulin resistance, not stablishing a proper hypothesis and failing to convey the real interest and objective of the studies. Similarly, the discussion is not focused and needs to be thoroughly revised.
Response: We appreciate your thoughtful review and helpful comments. We apologize for the inappropriate and confusing text. As you indicated, we have made appropriate corrections to the introduction of the abstract. (P1 L11).
Of crucial importance, the actual study design is not clear, since there is no information on when the test beverages were consumed by the volunteers. Together with the high-fat, high-carbohydrate cookies? Before? After? If so, with how much time difference? How much time did the volunteers have to ingest the cookie (and beverage)? This is important since incretins’ response occurs fast after food consumption and therefore the amount of food and the time available for its intake must be standardized in both studies. Could this have affected the differences observed in study 2 that were not found in study 1? Also, please indicate how much apart was each intervention (you indicate that after a 1-week run-in period in line 126).
Response: Thank you for pointing this out. This is important for the reproducibility of the test. We have clearly noted the conditions of consumption of the test beverages and cookies and the duration of each intervention (P3 L92 and P4 L165).
Other important information missing in the manuscript is the inclusion criteria or which parameter was used to estimate the sample size, among others. At this respect, it is not sufficient to indicate that sample size was estimated according to a non-referred preliminary study (line 119). Was post-prandial GLP-1 concentration (primary outcome) used for calculation of the sample size? Also, please justify the selection of GLP-1 levels as primary outcome.
Response: Thank you for your kind review and informative comments. We have corrected the error in the sample size estimation section and apologize for the error. The sample size was estimated using the primary endpoint, GLP-1 (P4 L147).
The fact that both studies were carried out in healthy men is an important drawback, as authors already acknowledged identifying it as a limitation. They should justify their selection of the study population.
Response: Thank you for your kind review and informative comments. As you pointed out, we did not include enough information on the selection of subjects. We have added this point to the discussion (P14 L433).
Among the contradictions found in the manuscript outstand the caffeine content of the test and placebo beverages. There are no two statements either in the abstract, experimental section, results (table 1) or discussion with the same information. Authors must clearly state how much caffeine contained each beverage in both studies. This incongruencies diminishes the confidence of readers on the manuscript.
Response: As you indicated, there was an error in the caffeine amount. It has been corrected. Thank you very much.
Authors indicate (lines 342-346) that the effect of the combined intake of GTC and CCA on incretins suggests that “the combination stimulates the digestion and absorption of nutrients, such as carbohydrates and lipids”. This is against the well-known effect of phenolic compounds inhibiting digestive enzymes and interfering on nutrients absorption. It is an affirmation not substantiated by any mechanistic study nor even a simple in vitro digestion of, for example, the test cookie with the combined GTC+CCA. But this suggestion is not even sustained by the results reported in the manuscript, since a reduced postprandial glucose levels were observed after the intake of the GTC+CCA. Again, important contradictions showing a poorly mediated discussion that weakens the relevance and potential impact of the manuscript.
Response: Thank you for your kind review. We would like to correct and apologize for the error in the discussion section, which did not make sense. We have clarified the description as follows (P13 L403).
It was suggested that the combined intake of GTC and CCA inhibits digestion and absorption of nutrients such as carbohydrates and lipids, and promotes increased secretion of GLP-1 and inhibits increased secretion of GIP, rather than directly stimulating K and L cells in the small intestine.
Other comments
In lines 84-90 it is said that “an optimal” dose of GTC and CCA were used, “reported as the minimum effective dose for metabolic disorders”. Is really a “minimum dose” the “optimal”? This is confusing.
Response: As you indicated, we have corrected the text to avoid misunderstanding (P2 L109 and L114).
The use of terms like test meal, test cookies, test beverages throughout the manuscript is confusing. Please standardize the terminology. Similarly, the terms high GTC/high CCA, low GTC/high CCA, high GTC/low CCA, high GTC/high CCA are not readers’ friendly. Could you not find some other easier way to refer to the different groups?
Response: We apologize for the confusion in terminology. We have unified the words "cookie meal" and "test beverage." Each test beverage was indicated as "Placebo" and "DoseA (B, C, D)".
In line 101 you mention that study diagrams are shown in Figures 1 and 2, but these figures correspond to results.
Response: We apologize for any errors in the text. We have deleted it (P3 L127).
In line 153 you indicate that the test beverages were analyzed by HPLC. Please briefly describe the analytical conditions. How was caffeine analyzed?
Response: We sincerely thank you. The method for measuring caffeine and the measurement conditions of HPLC have been added (P5 L188).
Next, you indicate that “In study 1, the GTC comprised catechin…”. Actually, in both studies the GTC and CCA had identical phenolic composition, only varying the final amounts of polyphenols. Most importantly, please clarify wat was used to prepare the GTC+CCA beverages, whether you used tea and coffee phenolic extracts or pure phenolic compounds. If using extracts, please describe how the extracts were obtained or, if using pure standards, indicate their precedence.
Response: Thank you for pointing out this important point. We have added a clarification that the test beverages were prepared using tea or coffee bean extract (P5 L194 and L195).
Please indicate in the experimental section how many high-fat, high-carbohydrate cookies (or grams) were consumed in each test day and specify the amount of glucose they provide (this is only mentioned in the abstract and discussion). Also, specify the volume of beverage consumed (this information is only included as a footnote in table 1).
Response: Thank you for pointing out an important point. We have corrected each of them for clarity (P1 L18, P4 L166, P6 L210 and P14 L453).
There is no information on how glucose and insulin were measured. Please include in section 2.5.
Response: Thank you very much. We have appended the details to the Methods section (P6 L221).
In line 234 you indicate that the high GTC/low CCA group exhibited a decreasing trend in the tAUC for serum insulin; a p=0.048 shows statistical significance, not “a trend”. On the contrary, in Table 3 p values of 0.058 and 0.065 for glucose and GLP-1, respectively, are not statistically significant, and therefore no asterisk should be included nor mention of significant differences compared to placebo.
Response: Thank you for your kind review. The comparison between the three groups shows a significant difference of P=0.048. And the comparison between the high GTC/low CCA group and the placebo group showed a decreasing “trend”, which was removed in this study.
In paragraph in lines 284-290 all tAUC should be changed to iAUC.
Response: We appreciate your kind review and have clarified the description as follows (P10 L324).
We assessed the differences in the iAUC for incretin responses because the fasting values fluctuated between each test. The tAUC was used for the blood glucose and serum insulin levels because the fasting values between each test were strictly regulated within individuals, and thus, postprandial values could be less than 0.
In line 379 authors refer to differences in estimated energy expenditure the day before the test between studies 1 and 2 as a potential explanation for the differences observed in fasting blood glucose levels. How the energy expenditure was measured/calculated must be described in the experimental section and results included in the manuscript to sustain this explanation.
Response: Thank you for your kind review. We would like to correct and apologize for the error in the discussion section, which did not make sense. We have corrected it from energy expenditure to energy intake. (P14 L447).
Finally, the post-prandial increase in blood glucose and insulin was not “suppressed” after intake of the GTC+CCA beverages, as concluded in line 384. Please rephrase.
Response: We appreciate your thoughtful review and helpful comments. As you indicated, We have modified the wording appropriately (P14 L451).
-------------------------
Sincerely yours,
Aya Yanagimoto
Biological Science Research Labs.
Kao Corporation
2-1-3, Bunka, Sumida-ku,
Tokyo, 131-8501, Japan
Telephone: +81-3-5630-7476
Fax: +81-3-5630-7456
E-mail: yanagimoto.aya@kao.com

Reviewer 2 Report (New Reviewer)
The paper is nice, well performed and advance our knowledge related to the effect of phytochemicals on decreasing risk of diabetes incidence. However, the manuscript in its current form requires some minor corrections:
-Abstract line,11-12. Authors have to change this statement sothat the effects of green tea and coffee come as suggestion rather than a fact.
- Introduction line, 35-38. Please fix the sentence. A general advice, try to avoid long sentences as they are difficult to follow.
Line 52. Which have different mechanisms of action. delete this part.
line 59-62. Split the sentence.
MM line 103, 104,113,114. delete.
line 157, 160. Change the word test into group or tested group.
line 172. change the subtitle to Biochemical analysis.
line 182. List the used softwares first.
Tables 3, 4 can be changed into bar charts. If authors want to keep current format, add vs. PLA between columns 3 and 4 and 4 and 5.
line 321. delete the word however.
line 324-325. fix the sentence.
line 366. the study limitations are well-acknowledged by authors.
Author Response
Response to Reviewer Comments
January 23, 2023
Nutrients
Dear Reviewers and Editors,
We would like to thank you very much for reviewing and commenting on our manuscript. We have carefully considered all your comments and made major revisions to the manuscript as a result. Below, we present our responses to each of the comments. We have significantly revised this manuscript in accordance with the your comments. We believe that these changes have greatly improved our paper. We hope that our revised version satisfies the reviewers and editors.
-------------------------
Reviewer comments:
The paper is nice, well performed and advance our knowledge related to the effect of phytochemicals on decreasing risk of diabetes incidence. However, the manuscript in its current form requires some minor corrections:
-Abstract line,11-12. Authors have to change this statement sothat the effects of green tea and coffee come as suggestion rather than a fact.
Response: We appreciate your thoughtful review and helpful comments. We apologize for the inappropriate and confusing text. As you indicated, we have made appropriate corrections to the abstract (P1 L11).
- Introduction line, 35-38. Please fix the sentence. A general advice, try to avoid long sentences as they are difficult to follow.
Response: Thank you for the pertinent comments. And we have divided long sentences appropriately (P2 L50).
Line 52. Which have different mechanisms of action. delete this part.
Response: We appreciate your comments and have removed the pertinent section (P2 L70).
line 59-62. Split the sentence.
Response: Thank you for the comments and we have divided long sentences (P2 L78).
MM line 103, 104,113,114. delete.
Response: We appreciate your comments and have removed the pertinent sentences (P3 L129, P3 L139).
line 157, 160. Change the word test into group or tested group.
Response: Thank you for your suggestion. I have changed the text appropriately, as you indicated (P5 L196 and 201).
line 172. change the subtitle to Biochemical analysis.
Response: We appreciate your thoughtful review and helpful comments. As you noted, we have corrected the subtitle (P6 L214).
line 182. List the used softwares first.
Response: Thank you for your kind review. As you indicated, we have rearranged the order of the method section (P6 L210).
Tables 3, 4 can be changed into bar charts. If authors want to keep current format, add vs. PLA between columns 3 and 4 and 4 and 5.
Response: Thanks to your important point, we have added "vs. Placebo" (Table 3 and Table 4).
line 321. delete the word however.
Response: We appreciate your comments and have removed the pertinent sentences (P13 L382).
line 324-325. fix the sentence.
Response: We appreciate your helpful comments. This sentence was overstated and has been removed (P13 L385).
-------------------------
It has been corrected correctly.Sincerely yours,
Aya Yanagimoto
Biological Science Research Labs.
Kao Corporation
2-1-3, Bunka, Sumida-ku,
Tokyo, 131-8501, Japan
Telephone: +81-3-5630-7476
Fax: +81-3-5630-7456
E-mail: yanagimoto.aya@kao.com

Round 2
Reviewer 1 Report (New Reviewer)
Authors have made an effort to improve the manuscript. However, this reviewer is not fully satisfied with the revised manuscript and the answers given to the initial comments comments and suggestions. Important information is still missing, such as how the beverages were prepared, clarifying whether tea/coffee phenolic extracts were used and, if so, how they were obtained and mixed. Perhaps this is sensible information that cannot be disclosed due to industrial interests of Kao Corporation; if so, this should be clearly stated in the manuscript and in the conflict of interests' disclosure.
In spite of this, the manuscript might be accepted for publication.
This manuscript is a resubmission of an earlier submission. The following is a list of the peer review reports and author responses from that submission.
Round 1
Reviewer 1 Report
I congratulate the author for fascinating and innovative studies, but there are some major issues to solve before publication, here is a list of these problems:
1)Line 30 the postprandial glucose level is influenced even by the level of catecholamine and cortisol, in stress situations, so this is taken into consideration in the introduction of the issue of DMT2
2)Line 50, Please detail the Insulinogenic index, and insert the formula.
3)line 54, typo incretins?
4)Line 85, please briefly,detail better what kind of beverage was used in studies 1 and 2 (line 89)
5)line 95-97; please register the study at www.clinicaltrial.gov and not only in the local network
6)line 101 why you use only men, provide a scientifical explanation in the text
7)Line 136, why in study 2 there is a substantial change in meal-time pattern, this influences the possible results, explain the reason for this choice.
8)line 137, explain that body fat measurement was assessed by BIA and not by the gold standard BOD-POD or DXA. and include this in the limitation of the studies.
9)table 1, what analytical method was used for the analysis of the beverages? LC-MS? HPLC? this information is pivotal for this study.
Please write this info in paragraph 2.4 and in table 1.
10)line 200, so you use ITT analysis of data of study 2? insert the proper term.
11)line 203, ?related hormone'? write Insulin in table 2 there is only this hormone.
12)line 316, various phytochemicals etc..., the authors could be more specific and insert a list of these compounds.
13)line 327 explains the K cell better.
14) In line 363, the test meal is not only high-fat but maybe high in sugars, have you ever dosed the simple carbohydrates in the test meal?
I belive that you could in consideration even split the 2 studies into 2 manuscripts if the author would, but this is not pivotal.
Best regards,
the referee
Author Response
Response to Reviewer Comments
December 11, 2022
Nutrients
Dear Reviewers and Editors,
We would like to thank you very much for reviewing and commenting on our manuscript. We have carefully considered all your comments and made major revisions to the manuscript as a result. Below, we present our responses to each of the comments. We have significantly revised this manuscript in accordance with the your comments. We believe that these changes have greatly improved our paper. We hope that our revised version satisfies the reviewers and editors.
-------------------------
Reviwer comments:
1)Line 30 the postprandial glucose level is influenced even by the level of catecholamine and cortisol, in stress situations, so this is taken into consideration in the introduction of the issue of DMT2
Response 1: We appreciate your thoughtful review and helpful comments. As you indicated, We have modified the wording appropriately regarding blood glucose regulation (P1 L33-36).
2)Line 50, Please detail the Insulinogenic index, and insert the formula.
Response 2: Thank you for thoughtful suggestions. As you point out, we have changed the description in abstract and manuscript as follows (P2 L61-62).
3)line 54, typo incretins?
Response 3: Thank you for your comment we have corrected our mistake (P2 L67).
4)Line 85, please briefly,detail better what kind of beverage was used in studies 1 and 2 (line 89)
Response 4: Thank you very much. We have appended the details (P3 L102-111).
5)line 95-97; please register the study at www.clinicaltrial.gov and not only in the local network
Response 5: We appreciate your comments and UMIN is not a local network, but a global network. We list below some papers that have been accepted by Nutrients and are registered in UMIN.
Unno, Keiko, et al. "Improvement of Depressed Mood with Green Tea Intake." Nutrients 14.14 (2022): 2949.
Kano, Toshiki, et al. "Impact of Transferrin Saturation and Anemia on Radial Artery Calcification in Patients with End-Stage Kidney Disease." Nutrients 14.20 (2022): 4269.
Togashi, Shintaro, et al. "Sensitivity and Specificity of Body Mass Index for Sarcopenic Dysphagia Diagnosis among Patients with Dysphagia: A Multi-Center Cross-Sectional Study." Nutrients 14.21 (2022): 4494.
Oshima, Shunji, Sachie Shiiya, and Yasuhito Kato. "Slow Drinking of Beer Attenuates Subjective Sedative Feeling in Healthy Volunteers: A Randomized Crossover Pilot Study." Nutrients 14.21 (2022): 4502.
6)line 101 why you use only men, provide a scientifical explanation in the text
Response 6: Thank you for your important comments. As you mentioned, we included men in this study because menstrual cycles may affect glucose metabolism in women. We have added this point to the limitations (P14 L412-414).
7)Line 136, why in study 2 there is a substantial change in meal-time pattern, this influences the possible results, explain the reason for this choice.
Response 7: We appreciate your thoughtful review and helpful comments. As you mentioned, due to differences in estimated energy expenditure on the day before the test between Study 1 and Study 2, fasting blood glucose levels on the day of the test may be higher in Study 1. Therefore, the same diet was consumed in each study, although caution must be exercised in interpretation. I have added this point to the limitations (P14 L414-418).
8)line 137, explain that body fat measurement was assessed by BIA and not by the gold standard BOD-POD or DXA. and include this in the limitation of the studies.
Response 8: Thank you very much. As you indicated, the BIA method was selected for this study because it is less burdensome to subjects. In addition, body fat percentage was measured at baseline and is unlikely to be involved in blood glucose or insulin. We have made an addition to the Methods section (P5 L165).
9)table 1, what analytical method was used for the analysis of the beverages? LC-MS? HPLC? this information is pivotal for this study.
Please write this info in paragraph 2.4 and in table 1.
Response 9: Thank you very much. We have appended the details (P6 L182-183 and P7 L198-199).
10)line 200, so you use ITT analysis of data of study 2? insert the proper term.
Response 10: We appreciate your thoughtful review and we have clarified our description (P8 L235-236).
11)line 203, ?related hormone'? write Insulin in table 2 there is only this hormone.
Response 11: Thank you for your kind review. We have corrected the points you raised (P8 L237 and P8 L243).
12)line 316, various phytochemicals etc..., the authors could be more specific and insert a list of these compounds.
Response 12: Thank you very much. We have listed the compounds as you have indicated (P13 L367-368).
13)line 327 explains the K cell better.
Response 13: We appreciate your helpful comment and we have corrected the points you raised (P14 L384).
14) In line 363, the test meal is not only high-fat but maybe high in sugars, have you ever dosed the simple carbohydrates in the test meal?
Response 13: Thank you for your comments on this new perspective. We have not done a study with a high-carbohydrate diet, but have modified the text to make the conditions clearer (P14 L431).
-------------------------
Sincerely yours,
Aya Yanagimoto
Biological Science Research Labs.
Kao Corporation
2-1-3, Bunka, Sumida-ku,
Tokyo, 131-8501, Japan
Telephone: +81-3-5630-7476
Fax: +81-3-5630-7456
E-mail: yanagimoto.aya@kao.com

Reviewer 2 Report
Introduction
I am missing the quote "These reports suggest that the GTC and CCA, which have different mechanisms of action, may have the potential to enhance the improvement of glucose metabolism. Furthermore, regulation of incretin is a potential therapeutic treatment option for insulin resistance, in addition to the control of glucose and insulin responses. Few human studies have evaluated the combined effects of GTC and CCA on postprandial glucose metabolism and gastrointestinal hormone secretion. Furthermore, the acute effects of their simultaneous consumption have not been clarified, and the mechanisms by which GTC and CCA affect glucose metabolism remain unclear. There are also no studies evaluating human glucose metabolism using different GTC and CCA doses; clarifying the minimum effective doses of GTC and CCA to better understand the relationship could provide valuable information for future study designs. "
lines 59-60: should not say "there are no studies" but "to the best of our knowledge there are no studies"
methodology
-how long did the study take? The given time range is not very specific.
- on what basis were incorrect eating habits identified?
-how many ml of these teas/coffees did they drink per day? You write something about a drink that contained the tested compounds, but I don't know what drink it is.
Author Response
Response to Reviewer Comments
December 11, 2022
Nutrients
Dear Reviewers and Editors,
We would like to thank you very much for reviewing and commenting on our manuscript. We have carefully considered all your comments and made major revisions to the manuscript as a result. Below, we present our responses to each of the comments. We have significantly revised this manuscript in accordance with the your comments. We believe that these changes have greatly improved our paper. We hope that our revised version satisfies the reviewers and editors.
-------------------------
Reviewer comments:
Point 1: I am missing the quote "These reports suggest that the GTC and CCA, which have different mechanisms of action, may have the potential to enhance the improvement of glucose metabolism. Furthermore, regulation of incretin is a potential therapeutic treatment option for insulin resistance, in addition to the control of glucose and insulin responses. Few human studies have evaluated the combined effects of GTC and CCA on postprandial glucose metabolism and gastrointestinal hormone secretion. Furthermore, the acute effects of their simultaneous consumption have not been clarified, and the mechanisms by which GTC and CCA affect glucose metabolism remain unclear. There are also no studies evaluating human glucose metabolism using different GTC and CCA doses; clarifying the minimum effective doses of GTC and CCA to better understand the relationship could provide valuable information for future study designs. "
Response 1: We appreciate your thoughtful review and helpful comments. As you pointed out, we have added a citation as per your instructions. (P2 L68-70).
Point 2: lines 59-60: should not say "there are no studies" but "to the best of our knowledge there are no studies"
Response 2: We appreciate your thoughtful review and helpful comments. As you pointed out, we may not have exhausted all the literature, so we have modified our wording appropriately (P2 L73-76).
Point 3: -how long did the study take? The given time range is not very specific.
Response 3: Thank you for thoughtful suggestions. We have appended the details of the test period (P3 L124-125 and L137).
Point 4: - on what basis were incorrect eating habits identified?
Response 4: We appreciate your thoughtful review and helpful comments. As you indicated, we have removed the following statement because an accurate dietary intake questionnaire is needed to point out incorrect dietary habits(P3 L131).
Point 5:
-how many ml of these teas/coffees did they drink per day? You write something about a drink that contained the tested compounds, but I don't know what drink it is.
Response 5: Thank you for your suggestion. As noted on P6 L181, 350 mL beverages were used in this Studies. To clarify this, we added the total volume consumed to Table 1.
-------------------------
Sincerely yours,
Aya Yanagimoto
Biological Science Research Labs.
Kao Corporation
2-1-3, Bunka, Sumida-ku,
Tokyo, 131-8501, Japan
Telephone: +81-3-5630-7476
Fax: +81-3-5630-7456
E-mail: yanagimoto.aya@kao.com

Round 2
Reviewer 1 Report
The issues of the first revision is corrected brightly.So I don't have any comment to the work, the work would be published on Nutrients.
Congratulation to all research team.